# COVID-19 in Patients with Hematologic Diseases

**DOI:** 10.3390/biomedicines10123069

**Published:** 2022-11-29

**Authors:** Ilaria Carola Casetti, Oscar Borsani, Elisa Rumi

**Affiliations:** 1Department of Molecular Medicine, University of Pavia, 27100 Pavia, Italy; 2Division of Hematology, Fondazione IRCCS Policlinico San Matteo, 27100 Pavia, Italy

**Keywords:** hematologic malignancies, COVID-19, autoimmune cytopenia, immune thrombocytopenia, vaccines, vaccine induced thrombotic thrombocytopenia

## Abstract

The COVID-19 outbreak had a strong impact on people’s lives all over the world. Patients with hematologic diseases have been heavily affected by the pandemic, because their immune system may be compromised due to anti-cancer or immunosuppressive therapies and because diagnosis and treatment of their baseline conditions were delayed during lockdowns. Hematologic malignancies emerged very soon as risk factors for severe COVID-19 infection, increasing the mortality rate. SARS-CoV2 can also induce or exacerbate immune-mediated cytopenias, such as autoimmune hemolytic anemias, complement-mediated anemias, and immune thrombocytopenia. Active immunization with vaccines has been shown to be the best prophylaxis of severe COVID-19 in hematologic patients. However, the immune response to vaccines may be significantly impaired, especially in those receiving anti-CD20 monoclonal antibodies or immunosuppressive agents. Recently, antiviral drugs and monoclonal antibodies have become available for pre-exposure and post-exposure prevention of severe COVID-19. As adverse events after vaccines are extremely rare, the cost–benefit ratio is largely in favor of vaccination, even in patients who might be non-responders; in the hematological setting, all patients should be considered at high risk of developing complications due to SARS-CoV2 infection and should be offered all the therapies aimed to prevent them.

## 1. Introduction

In December 2019, a pneumonia associated with a novel coronavirus (SARS-CoV2) was first reported to emerge in Wuhan, China [1,2]. The first descriptions appeared in the medical literature in early January 2020 and the new disease was named COVID-19. The outbreak was declared a pandemic in March 2020, due to the rapid diffusion of the infection throughout the world. Since December 2020, variants of concerns with a possible escape to prior immunization have been reported [3,4]; the most recent, named the Omicron variant, emerged at the end of 2021 and encodes a large number of genomic mutations, increasing transmissibility [4]. Until now, more than 6.5 million COVID-related deaths have been reported globally, while COVID-19 is evolving to an endemic disorder. The clinical course may be heterogeneous, ranging from asymptomatic infection to severe respiratory insufficiency requiring hospitalization. Compared with COVID-19 patients without cancer, patients with cancer had higher death rates, higher rates of intensive care unit (ICU) admission, higher probabilities of having severe or critical symptoms, and higher chances of needing invasive mechanical ventilation. Patients with hematologic malignancies (HMs) and lung cancer showed higher rates of severe events compared with patients without cancer [5]. Beyond HMs, hematologic autoimmune manifestations were also described in association with COVID-19, and patients with a previous diagnosis of autoimmune blood disorders, such as immune thrombocytopenia (ITP) and autoimmune hemolytic anemias (AIHA), frequently experienced exacerbation of their hematologic condition [6,7].

It has rapidly become clear that COVID-19 has a more severe clinical course in HM patients, especially those receiving bone-marrow-suppressing agents or having comorbidities [8,9]. Many treatment strategies, new therapies, and vaccines have been developed and, almost one year after the diagnosis of the first case, the vaccination campaign started from individuals at high risk of complications due to the infection. On the other hand, hematologic patients may also have a reduced immune response to vaccines, due to impaired humoral response or ongoing chemotherapy [10].

We conducted a review of the literature about blood disorders, COVID-19 infection, and the SARS-CoV-2 vaccine by searching for indexed and published articles and abstracts until September 2022 in MEDLINE via PubMed. The following keywords were used: hematologic malignancies, SARS-CoV-2, COVID-19 infection, autoimmune hemolytic anemia (AHIA), immune thrombocytopenia (ITP), vaccine-induced thrombotic thrombocytopenia (VITT), and COVID-19 vaccines. We selected only articles including human subjects; a total of 58 articles involving patients with hematologic diseases and COVID-19 were included.

## 2. Impact of COVID-19 in Patients with Hematological Malignancies

### 2.1. Risk of COVID-19 in Patients with HM

Many reports underlined the frailty of neoplastic patients in the context of the pandemic. In a cohort study, data on patients with active or previous malignancy; aged 18 years and older; with COVID-19 from the USA, Canada, and Spain; from the COVID-19 and Cancer Consortium (CCC19) database were collected between 17 March and 16 April 2020. Independent factors associated with increased 30-day mortality were: age, male sex, smoking status, number of comorbidities, Eastern Cooperative Oncology Group (ECOG) performance status of 2 or higher, and active cancer. Patients with cancer and COVID-19 showed a high 30-day all-cause mortality with an impact of general risk factors, but also risk factors unique to neoplastic patients [11]. An update of the COVID-19 and Cancer Consortium (CCC19) confirmed the risk factors already highlighted and suggested that HM patients have worse outcomes compared to subjects with solid tumors. The study analyzed a total of 4966 patients (median age 66 years, 51% female, 50% non-Hispanic white); 2872 (58%) were hospitalized and 695 (14%) died; 61% had concomitant cancer (diagnosed or treated within the year prior to COVID-19 infection). Increased age, male sex, obesity, cardiovascular and lung comorbidities, diabetes mellitus, kidney disease, non-Hispanic black race, Hispanic ethnicity, worse ECOG performance status, recent cytotoxic chemotherapy, and HM were associated with higher COVID-19 severity. Among patients requiring hospitalization, thrombocytopenia, lymphopenia or lymphocytosis, increased absolute neutrophil count, increased creatinine, troponin, lactate dehydrogenase (LDH), and C-reactive protein were associated with higher COVID-19 severity. Patients diagnosed in the first phase of the COVID-19 pandemic (January–April 2020) had worse outcomes than those diagnosed later. Specific combinations of chemotherapeutic drugs (e.g., R-CHOP—Rituximab, cyclophosphamide, hydroxydaunorubicin hydrochloride (doxorubicin hydrochloride), vincristine (Oncovin), and prednisone—platinum and etoposide, and DNA methyltransferase inhibitors) were associated with high 30-day all-cause mortality [12].

Hematologic patients were reported to be particularly frail, both in the outpatient and inpatient context from the very beginning of the pandemic. Cohort studies evaluated many series of patients with HM and COVID-19, recognizing a few risk factors for an adverse outcome: age, comorbidities, active HM, type of HM, ICU stay, and the use of ventilation.

In a series of 1250 patients with myeloid neoplasms regularly followed at a public university hospital in Milan, Italy, 16 developed COVID-19 in a 6-week period (10 March–24 April 2020) and 5 died. COVID-19 prevalence (1.28%) was higher than that observed in Lombardy in the same period (0.5–0.7%), possibly indicating a higher susceptibility of myeloid outpatients versus hospitalized ones [13].

A paper from the King’s College of London reported a significantly higher rate of mortality in COVID-19-patients with hematological neoplasms, compared to patients with a negative hematological history. The type of underlying malignancy (lymphoid vs. myeloid) did not influence the outcome, but the treatment had a strongly negative impact on mortality, with a worse outcome in patients receiving therapy, compared with the age- and gender-matched general cohort. Moreover, detection of viral RNA in the nasopharyngeal swabs/bronchoalveolar lavage was much more prolonged in patients with hematological cancer [14].

An Italian retrospective/prospective observational study (ITA-HEMA-COV project, NCT04352556) included 536 patients with SARS-CoV-2 infection (COVID-19) and a history or active hematologic malignancies, adding evidence that patients with hematological malignancies have worse outcomes than both the general population with COVID-19 and patients with hematological malignancies without COVID-19 [15]. In this study, COVID-19 was associated with a mortality of 33% in patients with myeloproliferative neoplasms (MPN), with the highest mortality in subjects with primary myelofibrosis (PMF) and better outcomes in patients with essential thrombocythemia (ET) and polycythemia vera (PV).

A recent report investigated the correlation between MPN, COVID-19 severity, and the presence of neutralizing autoantibodies (AAbs) against type 1 IFN. [16] In the literature, at least 15% of patients with life-threatening COVID-19 pneumonia were shown to have AAbs against IFN-I, which precede SARS-CoV-2 infection [17,18]. The preliminary results highlighted a higher prevalence of AAbs to IFN-I in patients with MPN compared to the general population, an association with the *JAK2*-unmutated genotype, with ET diagnosis, and a potential higher rate of severe COVID-19, suggesting that AAb detection may enable early identification of patients who are likely to develop life-threatening COVID-19 in a vulnerable population, such as patients with HM.

The ITA-HEMA-COV project also evaluated patterns of seroconversion for SARS-CoV-2 IgG in patients with HM [19]. A total of 237 patients with SARS-CoV-2-documented infection with at least one SARS-CoV-2 IgG test performed were analyzed. Among these, 62 (26.2%) had myeloid, 121 (51.1%) had lymphoid, and 54 (22.8%) had plasma cell neoplasms. Overall, 164 patients out of 237 (69%) had detectable IgG SARS-CoV-2 serum antibodies. In addition, 73 patients out of 237 (31%) resulted as serologically negative with no significant difference between myeloid or lymphoid/plasma cell neoplasms. Chemoimmunotherapy was associated with a lower rate of seroconversion, even a long time after therapy discontinuation, suggesting that treatment-mediated immune dysfunction is the main cause of the higher mortality rate.

The COVID-19 pandemic has forced clinicians and patients to choose between pre-COVID-19 standards of care and altering care to reduce the exposure of people to COVID-19. This caused significant delays or the interruption of treatment in oncologic patients. A study from July 2020 suggested a significant decrease in the number of patients with cancer having encounters. Considering the month of April 2020 vs. April 2019, patients with pulmonary, colorectal cancer, or HM had their medical encounters reduced (−39.1%, −39.9%, and −39.1%, respectively) less than patients with breast cancer, prostate cancer, and melanoma (−47.7%, −49.1%, and −51.8%, respectively). In addition, cancer screenings decreased dramatically, especially for breast cancer (−89.2%) and colorectal cancer (−84.5%) [20].

The same warning about cancer screening arose in the UK, as most procedures were suspended, routine work delayed, and only urgent symptomatic cases prioritized for diagnostic and surgical intervention. A population-based modelling study estimated a significant increase in the number of avoidable cancer deaths as a result of diagnostic delays due to the COVID-19 pandemic in the UK [21].

A summary of studies on HM patients with COVID-19 is shown in Table 1.

### 2.2. Psychological Distress of Patients with HM during the Pandemic

The COVID-19 pandemic also affected mental health, especially for people already struggling against cancer. Although remote health technologies have addressed some of the medical needs, the mental health of these patients was underreported. An international study used a validated artificial intelligence framework to conduct a comprehensive real-time analysis of two datasets of 2,469,822 tweets and 21,800 discussions by patients with cancer during this pandemic: the most concerns were expressed regarding delayed diagnosis, missed or postponed treatments, and impaired immunity. All patients expressed significant negative feelings, with fear being the predominant emotion [22]. A longitudinal study surveyed the impact of the pandemic on patients with cancer receiving Patient Advocate Foundation services in the US. In a total of 1529 patients, fear of COVID-19 resulted to be associated to psychological distress and delays in care; patients who were personally involved had their cancer care even more delayed. Among 1199 respondents to the survey, 94% considered themselves high risk for COVID-19. Fear of COVID-19 was associated with a higher mean psychological distress. Additionally, 47% of the respondents reported delaying care, with respondents with more fear having higher percentages of delayed care than those with less (56 vs. 44%). For respondents with a COVID-19 diagnosis in their family, distress scores were comparable, even if delays in care were higher than those without COVID-19 (58% vs. 27%) [23].

Patients with HM, as other cancer patients, reported high levels of anxiety and depression during the COVID-19 pandemic, because they represent a high-risk population and because antineoplastic treatments were often delayed. At least one third of the respondents to an online survey conducted through established hematology groups reported clinical levels of distress and identified unmet needs. The main concerns were associated with fear of cancer recurrence among respondents in remission and low-income people, and younger age and women were more vulnerable to anxiety and post traumatic disorders [24].

The impact of the COVID-19 pandemic on HM patients is depicted in Figure 1.

## 3. Immune-Mediated Cytopenias Triggered by COVID-19 Infection

The development of autoimmune manifestations following viral infection is a well-described complication caused by multiple mechanisms. Some viruses, including SARS-CoV-2, have the ability to induce hyperstimulation of the immune response [25]. Molecular mimicry, i.e., antibodies against SARS-CoV-2 spike glycoproteins cross-reacting with structurally similar host peptide protein sequences, may also play an important role in this phenomenon [26]. As an additional mechanism, COVID-19 infection can activate the complement cascade and the coagulation, inducing inflammation [27]. Complement activation can induce complement-mediated hemolytic anemias. Mechanisms of immune-mediated cytopenia are depicted in Figure 2.

This immune activation can cause hematological manifestations, such as immune thrombocytopenia (ITP) and hemolytic anemias. ITP has been reported following SARS-CoV-2 infection in several case reports [28,29,30,31,32].

Thrombocytopenia has been frequently described in COVID patients, mostly with mildly reduced platelet count, while severe thrombocytopenia is rare. It can be caused by various factors: shortened survival time of platelets, reduction in their production from megakaryocytes due to inflammation cytokines release and the immune-mediated mechanism. Patients with ITP and COVID-19 were treated with steroids and intravenous immunoglobulin (IVIg) and experienced platelet count recovery, confirming the autoimmune mechanism underlying ITP.

The occurrence or exacerbation of complement-mediated hemolytic anemias has also been described in patients with COVID-19. Complement-mediated anemias include autoimmune hemolytic anemia (AIHA), cold agglutinin disease (CAD), paroxysmal nocturnal hemoglobinuria (PNH), and hemolytic uremic syndrome (HUS). All of them may experience flares upon several triggers, including viral infections. COVID-19 is also a possible trigger through complement pathway activation by viral proteins, molecular mimicry, and autoantibodies production.

Many cases of reactivation or first diagnosis of hemolytic anemias have been reported. In PNH, red blood cells (RBCs) are subject to complement-mediated destruction, due to the acquisition of a somatic mutation in the PIG-A (Phosphatidylinositol glycan biosynthesis class A protein) gene, resulting in the loss of surface proteins including the complement inhibitors CD55 and CD59. Most patients are successfully managed with complement inhibitors targeting C5 (eculizumab or ravalizumab). Hemolytic flares in PNH after COVID-19 infection had a different severity with some patients requiring additional doses of the complement inhibitors or RBC transfusions [33,34,35,36]. Most cases only experienced laboratory signs of hemolysis with a mild reduction of hemoglobin (Hb) levels [37]. However, in a recent survey on a large cohort of PNH patients (more than half of all Italian patients), PNH subjects, either on complement inhibition or not, did not show a significantly higher risk of SARS-CoV-2 infection compared with the general population [38].

Similarly, many cases of AIHA and CAD development or reactivation during COVID-19 infection were reported, most of them requiring RBC transfusion due to severe anemia and steroid treatment or Rituximab [6,39,40,41,42,43,44]. Anemia associated with increased LDH and other hemolytic markers may be observed during infections but autoimmune hemolysis should be suspected if a patient with COVID-19 shows unexplained or persistent anemia. Interestingly, in the literature, some cases of underlying B cell malignancies were reported after CAD onset. Again, molecular mimicry could be the most relevant factor in the development of SARS-CoV-2-induced AIHA. Immunological cross-reactivity between Ankyrin-1, an erythrocyte membrane protein, and the viral protein Spike has been hypothesized as contributing to the pathogenesis of AIHA in patients with COVID-19 [45].

Immune-mediated cytopenias are listed in Table 2.

## 4. Vaccination against COVID-19 and Response of HM Patients to Vaccines

At the end of 2020, a few effective vaccines against SARS-CoV-2 have become available [46,47,48,49]. From December 2020 through March 2021, the European Medicines Agency approved four vaccines on the basis of randomized, blinded, controlled trials: two messenger RNA-based vaccines—BNT162b2 (Pfizer–BioNTech) and mRNA-1273 (Moderna)—that encode the spike protein antigen of SARS-CoV-2, encapsulated in lipid nanoparticles; ChAdOx1 nCov-19 (AstraZeneca), a recombinant chimpanzee adenoviral vector encoding the spike glycoprotein of SARS-CoV-2; Ad26.COV2.S (Johnson & Johnson/Janssen), a recombinant adenovirus type 26 vector encoding the SARS-CoV-2 spike glycoprotein. More than 12 billion doses have been administered globally until now. There is a clear consensus that these vaccinations help in preventing hospitalizations and in decreasing mortality after SARS-CoV-2 infections. COVID-19 vaccination is, therefore, recommended in immunocompromised patients, hematologic patients included [50]. However, information on vaccine safety and efficacy for patients with HM is still limited, as most trials (e.g., the registrational trials for Moderna COVID-19 vaccine and ChAdOx1 nCoV-19/AZD1222- University of Oxford, AstraZeneca COVID-19 vaccine) did not include patients with cancer. Only the phase 3 trial of BNT162b2 (Pfizer-BioNTech COVID-19 vaccine) and of Ad26.COV2.S (Janssen/Johnson & JohnsonCOVID-19 vaccine) enrolled a small percentage of patients with cancer (4% and 0.5%, respectively).

Most anticancer therapies, especially anti-CD20 antibodies, may result in a prolonged depletion of healthy B cells and are, therefore, immunosuppression. This impairs the humoral response and patients may have a reduced or absent response to common vaccines. A study from 2009 showed that lymphoma patients receiving rituximab-containing regimens achieved an adequate immunological response to influenza A (H1N1) vaccination [51]. On the other hand, immune checkpoint (anti-PD-1) treatment may enhance the vaccination response, leading to adverse events. In a review from 2021, ten studies assessing the safety and efficacy of influenza vaccination in cancer patients receiving anti-PD-1 were evaluated. Most of them had solid cancer, but HM was also included. No severe vaccination-related adverse events were reported. The pooled incidence of grade 3–4 toxicities was 7.5%. Based on the results from this study, influenza vaccination was demonstrated to be safe and effective in cancer patients receiving immune checkpoint inhibitors [52].

Considering these data, a reduced response to the COVID-19 vaccine was also expected and the treatment may impact the response in HM patients. The UK PROSECO (Prospective observational study evaluating COVID-19 vaccine responses) study showed an undetectable humoral response following two vaccine doses in more than a half of patients with B-cell malignancies undergoing active anticancer treatment. Moreover, 60% of patients on antiCD20 therapy had undetectable antibodies following full vaccination within 12 months of receiving their treatment against cancer [10]. A recent study evaluated the seroconversion rates against SARS-CoV-2 spike protein after FDA-approved COVID-19 vaccines. Compared with solid tumors, a significantly lower rate of seroconversion was observed in patients with HM (98% vs. 85%, respectively), particularly recipients undergoing immunosuppressive therapies such as anti-CD20 monoclonal antibodies (70%) and stem cell transplantation (73%). On the other hand, patients receiving immune checkpoint inhibitor therapy showed high seroconversion after vaccination (97%) and patients with prior COVID-19 infection had higher anti-spike IgG titers post-vaccination [53].

Many studies confirmed the reduced response after SARS-CoV-2 vaccination in patients with HM, particularly in patients previously treated with anti-CD20 antibodies or Bruton’s tyrosine kinase inhibitors (BTKi), both in comparison with healthy subjects and to patients with solid tumors.

A large prospective study, the CAPTURE study (Coronavirus disease 2019 (COVID-19) antiviral response in a pan-tumor immune monitoring) evaluated 585 patients with cancer following administration of two doses of BNT162b2 or AZD1222 vaccines, administered 12 weeks apart. Seroconversion rates after two doses were 85% and 59% in patients with solid tumors and HM, respectively. By comparison with individuals without cancer, patients with hematological, but not solid, malignancies had reduced neutralizing antibody (NAb) responses and anti-CD20 treatment was associated with undetectable NAb titers (NAbT). Vaccine-induced T cell responses were detected in 80% of patients and were comparable between vaccines or cancer types [54].

Another study conducted in Italy assessed the serological response to the COVID-19 mRNA vaccine in cancer patients receiving chemotherapy compared with healthy controls, confirming that HM patients have a reduced response. A total of 195 cancer patients (169 with solid tumors and 26 with HM) and 400 randomly selected controls who had been administered a Pfizer-BioNTech or Moderna COVID-19 vaccine in two doses were compared. The threshold of positivity was 4.33 BAU/mL (Binding antibody units/mL). The seropositivity rate was lower in patients than controls (91% vs. 96%), with an age/gender-adjusted rate ratio of 0.95. Positivity was found in 97% of solid cancers, but only in 50% of HM [55].

A similar prospective study evaluated the seroconversion rates and anti-SARS-CoV-2 spike protein antibody titers following full vaccination with BNT162b2 or mRNA-1273 SARS-CoV-2 vaccines in patients with cancer from January to April 2021. Most patients (94%) achieved seroconversion, but seroconversion rates and anti-spike antibody titers in patients with HM were significantly lower than in those with solid tumors. Specific anti-SARS-CoV-2 antibodies were not detectable in any of the patients with a history of anti-CD-20 antibody in the 6 months before vaccination [56].

A recent meta-analysis included only patients with HM. In addition, 26 studies with control arms were evaluated. After the first dose of vaccination, patients with HM had significantly lower seroconversion rates than controls (33.3% and 74.9%, respectively), especially in chronic lymphocytic leukemia (CLL) patients and patients treated with anti-CD20 antibodies or BTKi. The seroconversion rates were demonstrated to be higher after the second dose, but with a significant difference persisting between these 2 groups (65.3% vs. 97.8%) [57].

An international observational study (Epidemiology of COVID-19 Infection in Patients with Hematological Malignancies: A European Haematology Association Survey- EPICOVIDEHA), aimed at evaluating the epidemiology and outcomes of patients with HM and COVID-19, collected data from 42 countries through an electronic database. The majority of HM patients who were infected were males (61.1%) and over 50 years of age (85.8%). Most of them (>80%) had underlying lymphoproliferative malignancies (mostly CLL, non-Hodgkin, and multiple myeloma) and (68.1%) received active treatment of underlying HM at the time of COVID-19 or within the prior 3 months. Most of the patients received an mRNA vaccine (BioNTech/Pfizer 69.9%, Moderna 17.7%), whereas the remaining 14 (12.4%) received a vector-based vaccine (AstraZeneca Oxford) or an inactivated vaccine (Sinovac CoronaVac); overall, the median time from the last dose of the vaccine and COVID-19 diagnosis was 64 days. Eighty-seven patients (77%) were considered fully vaccinated, whereas the remaining 26 received only 1 shot; in all fully vaccinated patients, COVID-19 was diagnosed more than 2 weeks after the second vaccine dose. Postvaccine IgG levels against the SARS-CoV-2 spike protein were analyzed in 40 (35.4%) fully vaccinated patients, 2 to 4 weeks from the last vaccine dose. Among these patients, only 13 (32.5%) presented an antibody response to the vaccine, whereas the remaining 27 (67.5%) were considered non-responders. Overall, 79 (60.4%) patients had a severe or critical infection and 75 (66.4%) required hospitalization; 16 (21.3%) of them were admitted to an ICU, and 10/16 required mechanical ventilation. The overall mortality rate was 12.4%. COVID-19 was the main or a secondary cause of death for all but 1 patient. No statistical difference in terms of mortality between partially or fully vaccinated patients (15.4% vs. 11.5%) and between patients achieving a serological response to the vaccine vs. non-responders (13.3% vs. 15.6%) was observed. The only factor independently related to the risk of death in the cohort of vaccinated patients was the age. Ten of 14 (71.4%) patients who died had underlying lymphoproliferative malignancies [58].

The EPICOVID-EHA update, recently published online, evaluated the breakthrough COVID-19 in HM patients. The survey included a total of 1548 cases, of whom 76% had lymphoid malignancies. The Omicron variant was prevalent (68.7%) among the cases that had the viral genome sequenced. A total of 91% of the patients received at least two vaccine doses before COVID-19, mostly mRNA-based (89%). Overall, 59% of the patients received specific treatment for COVID-19. The mortality after a 30-day follow-up was 9%, with a mortality rate in patients with the Omicron variant of 7.9%, comparable to that reported for the previous variants and significantly lower than in the period before vaccination (31%). In the univariable analysis, older age, active HM, and severe and critical COVID-19 were associated with mortality. Patients receiving monoclonal antibodies had a lower mortality rate, even when COVID-19 was severe or critical. In the multivariable model, older age, critical COVID-19, active HM, and at least 2–3 comorbidities were correlated with a higher mortality. Monoclonal antibodies, alone or combined with antivirals, were observed as protective, even in this particular setting. Mortality was significantly lower than in the pre-vaccination era, but COVID-19 in HM is still associated with considerable mortality and the role of monoclonal antibodies seems to be relevant to improve the outcome in patients with HM [59].

During the pandemic, a few viral new variants with an increased ability to escape vaccines were reported. The current dominating Omicron strain turned out to be the most resistant among the variants known to date. Bivalent mRNA-based vaccines are now becoming available in most countries and proved good efficacy against the Omicron variant without evident safety concerns [60].

## 5. Antiviral Therapies and Prevention of COVID-19 in Immunocompromised Patients

HM patients are at high risk of severe infection and may require prophylactic or early therapeutic interventions. Some antiviral drugs have been developed and are active against many variants of COVID-19, including the most recent Omicron variants: Nirmatrelvir/Ritonavir, Remdesivir, and Molnupiravir. Nirmatrelvir is an oral protease inhibitor and is usually administered in combination with Ritonavir, a drug already commonly used in HIV infection treatment. It has shown efficacy in high-risk, unvaccinated patients infected with the B.1.617.2 (Delta) variant [61] and in patients 65 years of age or older infected with the B.1.1.529 (Omicron) variant. No difference in hospitalization rate was observed in younger adults [62]. Remdesivir was the first antiviral drug approved for the treatment of COVID-19. The PINETREE trial showed that nonhospitalized patients who were at high risk for severe COVID-19 had a significant benefit from a 3-day administration of remdesivir, with an acceptable safety profile and decreased risk of hospitalization or death than placebo (−87%) [63]. Molnupiravir is an oral antiviral product that was demonstrated to be active against COVID-19 infection, reducing the risk of hospitalization or death in at-risk, unvaccinated adults with COVID-19 (MOVe-OUT trial) [64].

Monoclonal antibodies directed to the Spike protein have also become available for a post-exposure prevention of the severe infection: the first were Bamlanivimab plus Etesevimab [65] and Casirivimab plus Imdevimab [66]. Those combination antibodies neutralize the ability of the virus to bind and enter the infected cell and gave interesting results, especially in preventing hospitalization and death and reducing viral load in high-risk ambulatory patients. On the other hand, none of them showed a neutralizing effect on the most recent Omicron variants BA.1 and BA.2 [67]. Sotrovimab, an engineered human monoclonal antibody that neutralizes SARS-CoV-2 and other sarbecoviruses, including SARS-CoV-1 (responsible for the SARS outbreak in 2002), was tested in the COVID-19 Monoclonal Antibody Efficacy Trial–Intent to Care Early (COMET-ICE) in high-risk ambulatory patients with mild to moderate COVID-19 [68]. It has been approved for treatment of patients that do not require oxygen therapy but have an increased risk of progressing to severe infection. Sotrovimab retained activity against many variants of concern, including the alpha, beta, gamma, delta, and lambda variants and Omicron BA.1, but the activity against BA.2 has shown to be limited [67,69]. The most recent monoclonal antibody, Bebtelovimab, has a potent neutralizing activity against the BA.2 variant [69] and was authorized for emergency use by the FDA in the USA but is still not available in other countries.

The only pre-exposure prophylaxis for COVID-19 currently available is a combination of 2 long-acting antibodies, tixagevimab and cilgavimab (AZD7442), which has been authorized recently to prevent severe infection in immunocompromised patients or in subjects that cannot receive specific vaccination, e.g., bone marrow transplant recipients. The phase 3 double-blind placebo-controlled trial (PROVENT) showed that a single dose of AZD7442 significantly prevented COVID-19. Symptomatic COVID-19 occurred in 8 of 3441 participants (0.2%) in the AZD7442 group and in 17 of 1731 participants (1.0%) in the placebo group (relative risk reduction, 76.7%); extended follow-up at a median of 6 months showed a relative risk reduction of 82.8%. Five cases of severe or critical COVID-19 and two COVID-19-related deaths occurred, all in the placebo group [70]. Interestingly, AZD7442 showed activity against Omicron variants, including BA.2 [69].

The therapeutic and prophylactic options currently available and recommended for HM are listed in Table 3.

## 6. Autoimmune Blood Disorders Triggered by COVID Vaccine

Vaccination was shown to potentially cause hematological side-effects, including ITP, exacerbation of autoimmune hemolytic anemias, and the so-called vaccine-induced thrombotic thrombocytopenia (VITT), a prothrombotic syndrome.

The latter phenomenon was described in a few case reports in subjects receiving vaccination based on adenoviral vectors (ChAdOx1 nCoV-19/AZD1222 and Ad26.COV2.S vaccine). VITT is characterized by thrombosis, mostly in unusual sites, and thrombocytopenia, and clinically resembles severe heparin-induced thrombocytopenia (HIT), a prothrombotic disorder caused by platelet-activating antibodies that recognize multimolecular complexes between cationic platelet factor 4 (PF4) and anionic heparin [71]. The first reports of VITT were published in June 2021 and included 39 patients, 27 of whom were women, with a median age of 41 years. Beginning 5 to 24 days after vaccination with ChAdOx1 nCoV-19/AZD1222, the patients presented with one or more thrombotic events and were demonstrated to have high levels of antibodies to PF4-polyanion complexes; their serum showed strong reactivity on the PF4–heparin ELISA and variable degrees of platelet activation in the presence of buffer that was, in most cases, greatly enhanced in the presence of PF4, as commonly detected in HIT [72,73,74]. Those vaccinated patients did not receive any heparin to explain the subsequent occurrence of thrombosis and thrombocytopenia. It has been recognized that many triggers other than heparin, e.g., viral and bacterial infections, knee replacement, and polyanionic medications, can cause a prothrombotic disorder that strongly resembles heparin-induced thrombocytopenia on both clinical and serologic grounds. This pathologic condition is called autoimmune heparin-induced thrombocytopenia [75]. Subsequently, a very similar case was reported in a patient who received the Janssen COVID-19 vaccine (Ad26.COV2.S) [76]. VITT was then proposed to be a variant of this autoimmune disorder, triggered by an adenovirus-based vaccine. The European Medicines Agency (EMA) estimated the incidence of VITT after vaccination with ChAdOx1 nCoV-19 to be between 1 in 125,000 and 1 in 1 million. More than 80% of the patients in VITT reports were females, with an age ranging from 20 to 55 years. The incidence appears to be very low, but VITT raised concerns about adenovirus-based vaccines and led to suspension in some European countries, including Italy [77]. The risk of development of VITT or other relevant vaccine-related side-effects is nonetheless much lower than the thrombotic risk caused by the disease itself [78].

VITT management recommendations have been provided recently by the International Society on Thrombosis and Haemostasis (ISTH) [79]. Early diagnosis, considering thrombocytopenia and thrombosis in unusual locations together with detection of anti-PF4-autoantibodies, is important to allow prompt treatment and prevent fatalities. Treatment includes high-dose IVIg and direct thrombin inhibitors, such as argatroban or bivalirudin.

Vaccination has also been associated with reactivation of many autoimmune phenomena, including autoimmune blood disorders. Vaccines activate immune-mediated mechanisms, mostly molecular mimicry, that induce protective immunity but, as well as infections, they may trigger an autoimmune reaction that can lead to ITP. Although thrombocytopenia secondary to COVID-19 vaccination has not been reported as a frequent adverse event in the clinical trials, occasional thrombocytopenia in previously healthy recipients appeared when vaccines started being used widely in the population [80,81]. More than 90% of the subjects recovered completely after the usual first-line treatment. The association of COVID-19 vaccines with ITP was overall not a concern. A US study identified 15 cases of thrombocytopenia among 18,841,309 doses of the Pfizer-BioNTech vaccine and 13 cases among 16,260,102 doses of the Moderna vaccine, calculating an incidence rate of thrombocytopenia of 0.80 per million doses for both vaccines. As the annual incidence rate of ITP is 3.3 cases per 100,000 adults, the observed number of thrombocytopenias, including ITP, following mRNA COVID-19 vaccines is comparable to the number of ITP cases expected [82]. Exacerbation of ITP has also been described: in a case series evaluating 52 chronic ITP patients, thrombocytopenia reactivation was reported in 12% of them, occurring independently of remission status, treatment, and vaccine type [83]. A more recent study documented ITP exacerbation in about 14% of chronic ITP patients, with bleeding in 2% of the cases and no influence of the type of vaccine on the outcome [84]. Subjects with ITP subsequent to vaccination usually respond well to first-line therapy, both to corticosteroids and IVIg, and the outcome is favorable [81].

As other autoimmune phenomena, AIHA and CAD might also be triggered by COVID-19 vaccines. Even if AIHA has not been considered as an adverse event associated with COVID-19 vaccines, some cases have been reported after receiving m-RNA-based vaccination [85,86,87,88]. Management includes steroids and, more rarely, IVIg or Rituximab. Overall, post-vaccine AIHA flares appear to be manageable and the benefits of vaccination greatly outweigh the risks, as post-infection flares are usually more severe.

## 7. Conclusions

Patients with HM, both myeloid and lymphoid cancer, represent a vulnerable population. In addition to the immune dysregulation caused by neoplasms themselves, many patients receive therapy that causes profound immunosuppression or prolonged cytopenias. In addition, with a median age for most diseases in the mid-60s, many patients are elderly and have concomitant comorbidities that further contribute to a heightened risk of a severe course of COVID-19, with a high fatality rate.

Patients with HM show a short-lasting protective response after COVID-19 infection and are at risk of re-infection and are candidates to active immunization through vaccines. The cost–benefit ratio is largely in favor of vaccination, even in HM, who might be non-responders. Side-effects are very limited, mild, and manageable, while mortality is decreasing sharply in the vaccination era. A subgroup of patients with HM, especially those with lymphoid malignancies, may not respond adequately and, therefore, need early therapeutic measures to prevent severe infection. The intervention should consider the availability of the new antiviral drugs and monoclonal antibodies and the interactions between cancer treatment/immunosuppressive treatment and specific anti-COVID-19 therapies.

COVID-19-related autoimmune blood disorders might be challenging, especially in patients with underlying autoimmune cytopenia, who are more prone to develop exacerbation of their condition. ITP and AIHA occurring during COVID-19 infection are more severe, more often complicated, and require a greater therapeutic intervention as compared to those developing after COVID-19 vaccines. Patients with autoimmune blood disorders should then be vaccinated and monitored properly.

## Figures and Tables

**Figure 1 biomedicines-10-03069-f001:**
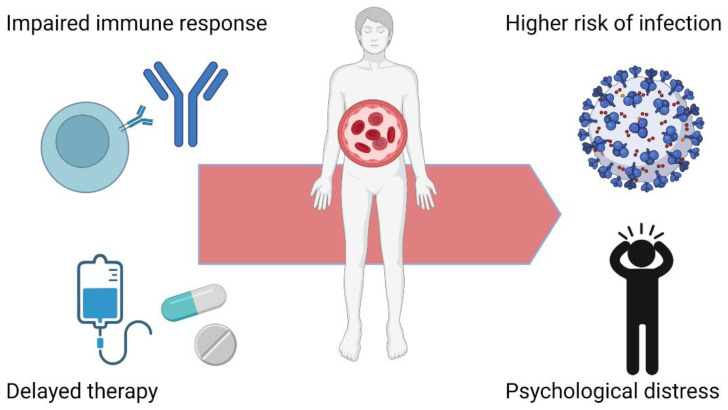
Impact of COVID-19 pandemic on HM patients. HM patients show an impaired immune response and suffered from delays of anticancer therapies during the pandemic. These conditions made them more vulnerable to infection and to develop psychological distress. Figure created with BioRender.

**Figure 2 biomedicines-10-03069-f002:**
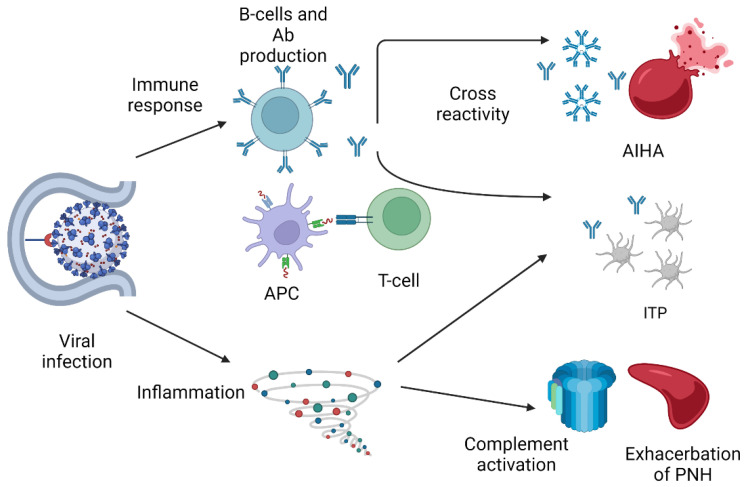
Mechanism of immune-mediated cytopenias triggered by COVID-19 infection. AIHA autoimmune hemolytic anemia, ITP immune thrombocytopenia, PNH paroxysmal nocturnal hemoglobinuria, Ab antibodies, and APC antigen-presenting cell. Figure created with BioRender.

**Table 1 biomedicines-10-03069-t001:** COVID-19 in patients with hematologic malignancies (HM).

Reference	Patient Population	Risk Factors	Clinical Severity	Mortality
[8]	128 pts with HM (13 with COVID-19) Median age 35 y	Co-bacterial infections	10 pts required oxygen	62%
[9]	35 pts with HM and COVID-19 Median age 69 y	Age, number of comorbidities		40%
[11]	928 pts with cancer and COVID-19 Median age 66 y	Age, male sex, smoking, number of comorbidities, ECOG > 2, active cancer	14% admitted to ICU 12% required ventilation	13%
[12]	4966 pts with cancer and COVID-19 (10% with HM) Median age 66 y	Age, male sex, obesity, CV and lung comorbidities, renal disease, diabetes mellitus, non-Hispanic black race, Hispanic ethnicity, worse ECOG, recent cytotoxic chemotherapy, HM	Hospitalization 58.7%	14%
[13]	1250 pts with HM (16 pts with COVID-19)	Age, number of comorbidities	Hospitalization 80% 12% admitted to ICU	30%
[14]	80 pts with HM and COVID-19 Mean age 69 y	Age, intensive therapy		39%
[15]	536 pts with HM and COVID-19 Median age 68 y	Age, AML diagnosis, indolent and aggressive NHL, PC neoplasms, severe COVID-19	15% admitted to ICU	36.9%
[16]	219 pts with MPN (29 pts with COVID-19) Median age 62 y	Presence of AAbs against type I IFN	Hospitalization 27.6%	

Pts: patients, ECOG: Eastern Cooperative Oncology Group; ICU: intensive care unit; CV cardiovascular; AML: acute myeloid leukemia; NHL non-Hodgkin lymphoma; PC plasma cell; MPN: myeloproliferative neoplasms; AAbs autoantibodies.

**Table 2 biomedicines-10-03069-t002:** Immune-mediated cytopenia triggered by COVID-19.

Immune Mediated Cytopenia	Reference	Median Hb/PLT Value at Nadir	N. of Patients	Management of the Cytopenia	Outcome
AIHA (warm and cold antibodies)	[36]	7 g/dL	7 (3 CAD)	Steroid, Rituximab, Transfusion	3 remission, 4 ongoing at the time of publication
	[37]	5.3 g/dL	1 (CAD)	Transfusion	Death
	[38]	1.6 g/dL	1 (mixed Abs)	Transfusion, steroid, Rituximab	Remission
	[39]	6.9 g/dL	1 (CAD)	Transfusion	Remission
	[40]	7.1 g/dL	2 (CAD)	Transfusion	Remission
	[41]	7.3 g/dL	3 (1 CAD)	Steroid, transfusion	Remission
ITP	[25]	2000/mmc	1	IVIg, steroid, transfusion, Rituximab	Remission
	[26]	20,000/mmc	2	Steroid	Remission
	[27]	17,000/mmc	3	IVIg	Remission
	[28]	9000/mmc	1	IVIg	Remission
	[29]	3000/mmc	3	Transfusion, IVIg, steroid	2 remissions, 1 death
PNH	[30]	6.5 g/dL	1	Eculizumab	Remission
	[31]	8.9 g/dL	1	Eculizumab	Remission
	[32]	9.8 g/dL	1	Steroid	Remission
	[33]	NA	4	Eculizumab, Ravulizumab, transfusion	3 remissions, 1 ongoing at the time of publication
	[34]	NA	3	Eculizumab	Remission
	[35]	NA	4	Eculizumab, Ravulizumab	Remission

AIHA: autoimmune hemolytic anemia, ITP: immune thrombocytopenic purpura, PNH: paroxysmal nocturnal hemoglobinuria, CAD: cold agglutinin disease; IVIg: intravenous immunoglobulin; Hb: hemoglobin; PLT: platelets; g/dL: grams per deciliter; mmc: cubic millimeter.

**Table 3 biomedicines-10-03069-t003:** Prophylactic and therapeutic options in patients with HM. Schedule of vaccines is reported based on the registration studies.

Pre-Exposure/Prophylactic Treatment	Schedule	Activity against Omicron Variant
Vaccines (most used in Western countries)		
mRNA-based		
BNT162b(Pfizer/Biontech) mRNA-1273 (Moderna)	2 doses 21 days 2 doses 28 days	Bivalent booster Bivalent booster
AdV-vectored		
ChAdOx1 nCov-19 (AstraZeneca/Oxford) Ad26.CoV2.S (Johnson& Johnson)	2 doses 4–12 weeks 1 dose	NA NA
Monoclonal antibodies		
Tixagevimab/Cilgavimab	150/150 mg iv day 1	BA.1 (reduced) and BA.2
Post-exposure/confirmed COVID-19 Treatment		
Antiviral therapy		
Remdesivir Nirmatrelvir + Ritonavir Molnupiravir	200 mg iv, day 1, 100 mg iv day 2–3 2 × 300/100 mg po for 5 days 2 × 800 mg po for 5 days	Yes Yes Yes
Monoclonal antibodies		
Etesevimab/Bamlanivimab Imdevimab/Casirivimab Sotrovimab Bebtelovimab (US only) Tixagevimab/Cilgavimab	1400/700 mg iv day 1 1200/1200 mg iv day 1 500 mg iv day 1 175 mg iv day 1 300/300 mg iv day 1	Markedly reduced Markedly reduced BA.1 only BA.1 and BA.2 BA.1 (reduced) and BA.2

iv: intravenous; po: per os.

## Data Availability

Not applicable.

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
