# Peer review of "COVID-19 in Patients with Hematologic Diseases"

_biomedicines, 2022, doi:10.3390/biomedicines10123069_

Round 1
Reviewer 1 Report
- Major comments:
Patients with hematologic malignancies (HM) are particularly vulnerable with regard to COVID-19. In the manuscript, Ilaria Carola Casetti et al. provided an overview of poor outcomes of COVID-19 patients with hematologic diseases, and highlighted the benefits of active immunization with vaccines for the best prophylaxis of severe COVID-19 in these patients. In addition, the review also pinpointed the recent developments of antiviral drugs and monoclonal antibodies that can be useful for the pre-exposure and post-exposure prevention of severe COVID-19 in patients with hematologic diseases.
As the virus evolves, the ongoing pandemic will remain a significant challenge for patients with hematologic malignancy. Therefore, understanding the pathophysiology and recognizing risk factors associated with poor prognosis are crucial in improving these patient outcomes with COVID-19 through up-to-date prophylaxis and therapeutic interventions.
- Major suggestions:
The risk factors of COVID-19 in patients with HM should be summarized and listed in Figure 1 or a separate table/figure for better illustration.
The review covers various topics and is relevant to the field. The review can be further improved by including the following suggestions listed in specific comments.
- Specific comments:
1) Line 9, “hardly affected”? Suggests using words like “heavily affected” here to avoid ambiguity.
2) Line 82, the full name of “R-CHOP” should be given. Rituximab, cyclophosphamide, hydroxydaunorubicin hydrochloride (doxorubicin hydrochloride), vincristine (Oncovin), and prednisone.
3) Line 96, here “in patients” should be referred to as “in COVID-19 patients”.
4) Line 178-179, typo of “SARS-Cov-2”
5) Figure 2, the full names of the abbreviations like “Ab” and “APC” should be listed in the Figure legend as well. Also, autoimmune hemolytic anemia (AIHA) also can be complement-mediated, according to the main text.
6) Line 208, the full name of “RBC” should be listed here for the first time. “red blood cell (RBC)”
7) Line 209, the full name of “PIG-A” should be listed. Phosphatidylinositol glycan biosynthesis class A protein (PIGA).
8) Line 214, “Hb levels” here referred as “Hemoglobin (HB) levels”?
9) Table 1, the full names of the abbreviations like “Hb”, “PLT”, “g/dl” and “mmc” should be listed in the Table legend as well.
10) Line 256, what does “ad” mean here?
11) Line 265, the full name of “PROSECO” should be listed. Prospective observational study evaluating COVID-19 vaccine responses.
12) Line 277, typo of “SARS_COV-2”
13) Line 279, “BTK inhibitors” should be referred as “Bruton's tyrosine kinase inhibitors (BTKi)”, as BTKi also appeared in Line 308.
14) Line 286, “NAbT” should be referred as “NAb titers (NAbT)”.
15) Line 294, “BAU/mL” here referred as “Binding antibody units (BAU)/mL”?
16) Line 302, “SARS antibodies” or “SARS-CoV-2 antibodies” here?
17) Line 316, the full name of “CLL” should be listed here, Chronic lymphocytic leukemia (also called CLL).
18) Table 2, the full names of the abbreviations like “iv” should be listed in the Table legend.
Reviewer 2 Report
The article presented to me for evaluation is a well-written literature review of SARS-Cov2 infections in patients with hematological diseases. The authors show the results of studies on the impact of infection on patients with hematologic malignancies, autoimmune-induced cytopenias, response to vaccination, prevention of infection, and occurrence of autoimmune processes after vaccination. The methodological side of the article doesn’t rise any objections, and the subject matter is important and actual - which is sure to interest the readers.
Please see my comments below.
Introduction:
Line 58: please add information how many human studies were included. I suggest to add prisma 2020 diagram.
Section 2.1:
Consider adding a table summarizing the results shown in discussed papers.
